# Digital Teaching Competence among Teachers of Different Educational Stages in Spain

Miguel Ángel García-Delgado *, Sonia Rodríguez-Cano, Vanesa Delgado-Benito and Cristina Di Giusto-Valle

Faculty of Education, Department of Education, University of Burgos, 09001 Burgos, Spain; srcano@ubu.es (S.R.-C.); vdelgado@ubu.es (V.D.-B.); cdi@ubu.es (C.D.G.-V.)
* Correspondence: mgd0111@alu.ubu.es

**Abstract:** The new educational reality requires teachers to have skills and competencies to improve the teaching-learning process and, therefore, the quality of teaching, integrating new technologies. To assess the competence level of teachers, a descriptive study was designed, in which 150 teachers from different stages in Spain took part and were administered the DigCompEdu Check-in questionnaire. Non-probabilistic (purposive) sampling was used. The results show an intermediate level of competence among teachers at all the educational stages surveyed. However, this level varies according to the stage at which they work, with secondary education, vocational training, GCE and university teachers standing out the most in the fields of competence analysed using the questionnaire. Different proposals for improvement are proposed too, as well as the existing coincidences with previous studies; furthermore, the need for training from the beginning of teacher preparation is highlighted, as well as the need for continuous training for active teachers to optimise the potential offered by the new technologies, also is very important that taking on challenges such as the correct communication trough technologies, the use and creation of digital content or the protection and security of online data, among others.

**Keywords:** teacher digital competence; primary education; secondary education; vocational education and training; university

## 1. Introduction

Technology has substantially changed how we obtain knowledge and information, communicate, work, and enjoy leisure time [1]. Along the same lines, some authors, such as Benavente-Vera et al. [2], stress the importance of new technologies in our daily lives, which have brought about significant changes in the world in which we live, and therefore consider it essential that everyone is trained to be digitally competent. Furthermore, digital competence helps try to solve and mitigate the challenges proposed by knowledge society as it allows each individual's cognitive, technical, and attitudinal skills to be revealed [3].

It should also be noted that new technologies have had a strong impact on education, and it is precisely at this point that we find different types of teachers; those who adapt easily, and those who resist, although the problem does not only lie in the latter, the transformation of teaching praxis is necessary to adapt it to the new reality derived from technological advances [4]. According to Cabero-Almenara and Palacios-Rodríguez [5], educational policies and scientific studies on education highlight the important role that teacher digital competence has acquired within the new educational reality in the classroom and the literacy process. Along these lines, Torres-Barzabal et al. [6] suggest that teachers are currently required to develop new competencies to carry out their work optimally. Therefore, they are expected to be able to incorporate technological knowledge and skills in the teaching-learning processes to improve teaching.

This gives rise to the term Teacher Digital Competence (CDD, as per its Spanish acronym). It does not refer solely and exclusively to training processes centred on the

instrumental use of technologies but covers a much broader spectrum [7]. Teacher Digital Competence involves not only the use of technologies within the teaching and learning process but also the environment in which experiences and learning take place and is aimed at maximising the possibilities offered by technologies to improve teaching practice [1].

In this sense, it is important to consider—from the field of Educational Technology—the need to review what elements are incorporated to continue advancing in studies and research in this field [8]. Likewise, authors such as Revelo Rosero et al. [9] highlight the need to master, use and innovate in the digital competence of teachers as an essential element for improving and promoting changes in education, thus enabling the acquisition of learning that allows the construction of knowledge.

Moreover, along the same lines, the need for teachers to train students to provide them with the tools to actively participate in social life and work in the new digital age is also pointed out [1]. Similarly, we must understand that the open curriculum approach is linked to the idea of not restricting or associating learning exclusively with the classroom environment but rather that learning takes place in different environments [10], and it is here that new technologies play an essential role.

For all these reasons, digital competence in teaching tends to be a recurrent goal for teachers and educational legislation. Although it is still not fully achieved, significant progress has been made [11]. Along the same lines, some authors [12] corroborate the growing boom and interest in Digital Competence in the field of research, with special importance being attached to the field of teaching in national research activity, as well as highlighting that the growing scientific production in this field will promote the creation of elements and improvements to favour the achievement of objectives in the field of teacher training.

For all the above reasons, national and supranational administrations have implemented various frameworks and tests for the assessment and accreditation of digital competence [13]. Based on the reality described in the Eurydice Report [14], there is a similar approach in all European countries to define teacher digital competence as a key element; moreover, it is a competence reflected in all stages of education and therefore involves all teachers. In this case, we have taken as a guide the Framework of Reference for Teacher Digital Competence [15], correlated with the Digital Competence Framework for Teachers (DigCompEdu) (2017) [1], but adapted to the characteristics of the Spanish education system, to categorise the levels of mastery of digital competence by teachers, organised by competences from level A1 to C2, establishing three differentiated levels: Basic [A1 (novice) and A2 (explorer)], Intermediate [B1 (integrator) and B2 (expert)] and Advanced [C1 (leader) and C2 (pioneer)].

Based on all of the above, the main objective of this study was to assess the perceived and real digital competence of teachers at different educational stages and in different parts of Spain to give a proxy picture of the reality of teachers in our education system and their potential shortcomings.

## 2. Materials and Methods

The research has a descriptive nature. Therefore, the objectives to which this article seeks to respond are the following:

- To assess the degree of teacher digital competence among teachers in a series of schools in Spain.
- To analyse the levels of digital competence among teachers at different stages of the education system.

To obtain data, the "DigCompEdu Check-In" questionnaire, described as a tool for teachers' self-reflection on digital issues, was used in its Spanish version [5], consisting of six fields, which are detailed below:

- Professional engagement: encompasses the competencies of organisational communication, professional collaboration, reflective practice and digital literacy.

- Digital resources: this covers the selection of resources, the creation and modification of resources and the administration, sharing and protection of content.
- Digital pedagogy: being able to teach, guide, and foster collaborative learning and self-directed learning.
- Assessment and feedback: developing assessment strategies, the ability to analyse evidence and evidence, and feedback and planning through digital technologies.
- Empowering students: favouring accessibility and inclusion of students, ensuring equal opportunities; differentiation and personalisation of tasks, adapting them to students' educational needs; and encouraging their active participation.
- Facilitating students' digital competence: fostering information and media literacy, promoting digital communication and collaboration, content creation, responsible use, and digital well-being and problem-solving in new technologies.

All items in the different fields are answered on a Likert-type scale with five response intervals, in which the participating teachers have to reflect on the extent to which they identify with the proposed statement [5]. In addition, socio-demographic data are collected: gender, age, years of experience, type of teaching situation (temporary, permanent position), educational stage, type of educational establishment, perceived socio-economic level of students, participation of the centre in digitalisation programmes, hours dedicated to the use of technology in the classroom, digital tools used for teaching, teachers' digital citizenship competence, participation in social networks and working conditions that favour the usage of digital technology.

The questionnaire also includes an evaluation scale to classify the teachers' level of competence according to their answers. According to Cabero-Almenara and Palacios-Rodríguez [5], there are six competence levels: basic (A1 and A2), intermediate (B1 and B2) and advanced (C1 and C2), which are characterised by the following features:

- Basic:
  - Beginner (A1): has little experience and little contact with educational technology and therefore requires guidance and training to improve their level of competence.
  - Explorer (A2): shows little contact with technologies and has not developed the necessary skills for inclusion in the classroom.

- Intermediate:
  - Integrator (B1): can use technologies and optimise resources to adapt them to different learning situations.
  - Expert (B2): this profile shows different strategies and skills that allow this type of teacher to continuously improve the use of digital tools according to the context in which they are used.

- Advanced:
  - Leader (C1): can serve as a guide for other teachers and can adapt the resources and knowledge they have access to suit the needs of their educational work.
  - Pioneer (C2): leads the innovation of technologies in their environment, has the reflective capacity to improve and question the different practices in their environment, and is a model for the other teachers.

The questionnaire was distributed electronically, requesting participation voluntarily and guaranteeing the protection of responses and their anonymity. Therefore, a non-probabilistic, purposeful sampling was carried out, in which one hundred and fifty teachers from different educational stages participated, based on the distribution stages of the Spanish educational system, and which was included as an additional item in the questionnaire for its subsequent classification.

As detailed above, the study sample comprises one hundred and fifty teachers from different educational stages (Table 1), of whom 60% are women and 40% are men.

**Table 1.** Distribution by educational stage according to gender.

| Educational Stage | Women | | Men | | Total | |
|---|---|---|---|---|---|---|
| | N | % | N | % | N | % |
| Pre-school and Primary Education | 34 | 22.7% | 12 | 8% | 46 | 30.7% |
| Secondary Education and Vocational Training | 36 | 17.3% | 16 | 10.7% | 42 | 28% |
| GCE and University | 30 | 20% | 32 | 21.3% | 62 | 41.3% |
| Total | 90 | 60% | 60 | 40% | 150 | 100% |

## 3. Results

The general data on Teacher Digital Competence after having carried out a descriptive statistical analysis of frequencies based on the assessment guidelines established in the questionnaire, grouped according to the educational stage in which the teachers in the sample carry out their teaching activity. On the other hand, as we can see in Table 2, we can see the level of competence of the teachers in relation to their experience and the educational stage in which they carry out their work. As we can see, teachers at higher educational stages have higher levels of competence, and in many cases, they are participants with extensive educational experience. Likewise, we observe that, in the lower competence levels, there is polarization, with teachers with little experience and a low competence level or with many years of teaching experience and a low level related to their level of digital competence; this does not occur among the teachers at GCE (General Certificate Education) and University, given that none of them appears at level A1.

**Table 2.** Competence level in relation to the educational stage and teaching experience.

| Educational Stage | Teaching Experience | Competence Level | | | | | | | | | | | | | |
|---|---|---|---|---|---|---|---|---|---|---|---|---|---|---|---|
| | | A1 | | A2 | | B1 | | B2 | | C1 | | C2 | | Total | |
| | | N | % | N | % | N | % | N | % | N | % | N | % | N | % |
| Pre-school and Primary Education | Between 1 & 5 years | 1 | 0.66% | 3 | 2% | 6 | 4% | 3 | 2% | 2 | 1.33% | 0 | 0% | 15 | 10% |
| | Between 6 & 10 years | 0 | 0% | 1 | 0.66% | 6 | 4% | 4 | 2.66% | 0 | 0% | 0 | 0% | 11 | 7.33% |
| | Between 11 & 15 years | 0 | 0% | 3 | 2% | 0 | 0% | 0 | 0% | 0 | 0% | 0 | 0% | 3 | 2% |
| | Between 16 & 20 years | 0 | 0% | 0 | 0% | 3 | 2% | 2 | 1.33% | 2 | 1.33% | 0 | 0% | 7 | 4.66% |
| | Over 20 years | 0 | 0% | 4 | 2.66% | 5 | 3.33% | 0 | 0% | 1 | 0.66% | 0 | 0% | 10 | 6.66% |
| Secondary Education and Vocational Training | Between 1 & 5 years | 0 | 0% | 3 | 2% | 2 | 1.33% | 5 | 3.33% | 0 | 0% | 0 | 0% | 10 | 6.66% |
| | Between 6 & 10 years | 0 | 0% | 0 | 0% | 2 | 1.33% | 2 | 1.33% | 0 | 0% | 0 | 0% | 4 | 2.66% |
| | Between 11 & 15 years | 0 | 0% | 0 | 0% | 0 | 0% | 1 | 0.66% | 3 | 2% | 0 | 0% | 4 | 2.66% |
| | Between 16 & 20 years | 0 | 0% | 0 | 0% | 1 | 0.66% | 2 | 1.33% | 0 | 0% | 0 | 0% | 3 | 2% |
| | Over 20 years | 2 | 1.33% | 1 | 0.66% | 9 | 6% | 3 | 2% | 4 | 2.66% | 2 | 1.33% | 21 | 14% |
| GCE and University | Between 1 & 5 years | 0 | 0% | 0 | 0% | 4 | 2.66% | 5 | 3.33% | 2 | 1.33% | 2 | 1.33% | 13 | 8.66% |
| | Between 6 & 10 years | 0 | 0% | 6 | 4% | 2 | 1.33% | 3 | 2% | 3 | 2% | 1 | 0.66% | 15 | 10% |
| | Between 11 & 15 years | 0 | 0% | 2 | 1.33% | 4 | 2.66% | 1 | 0.66% | 1 | 0.66% | 0 | 0% | 8 | 5.33% |
| | Between 16 & 20 years | 0 | 0% | 1 | 0.66% | 1 | 0.66% | 2 | 1.33% | 0 | 0% | 0 | 0% | 4 | 2.66% |
| | Over 20 years | 0 | 0% | 0 | 0% | 9 | 6% | 7 | 4.66% | 5 | 3.33% | 1 | 0.66% | 22 | 14.66% |
| Total | | 3 | 2% | 24 | 16% | 54 | 36% | 40 | 26.66% | 23 | 15.30% | 6 | 4% | 150 | 100% |

As we can see in Figure 1, the results show a clear difference in the level of competence between GCE and university teachers with respect to the other groups, with the difference particularly evident at the highest levels (C1 and C2). Likewise, we find that regardless of the stage in which their activity is framed, teachers show an intermediate-high level of competence. However, focusing on the groups analysed, preschool and primary school teachers reach most level C1, although the majority are represented between levels A2 and B1, with a minority at level A1. If we refer to secondary education and vocational training teachers, we find a minority is represented at the lowest level, as well as at the highest level (C2); in general, most of the population of this educational stage is framed in the intermediate level (B1 and B2), with a population also at A2 and C1, although less

represented than in the intermediate levels. Finally, with regard to secondary school and university teachers, most of them appear in the intermediate and high levels (B1 to C1). Notably, they are not represented at the entry-level (A1), and they are the most represented population at the highest competence level (C2).

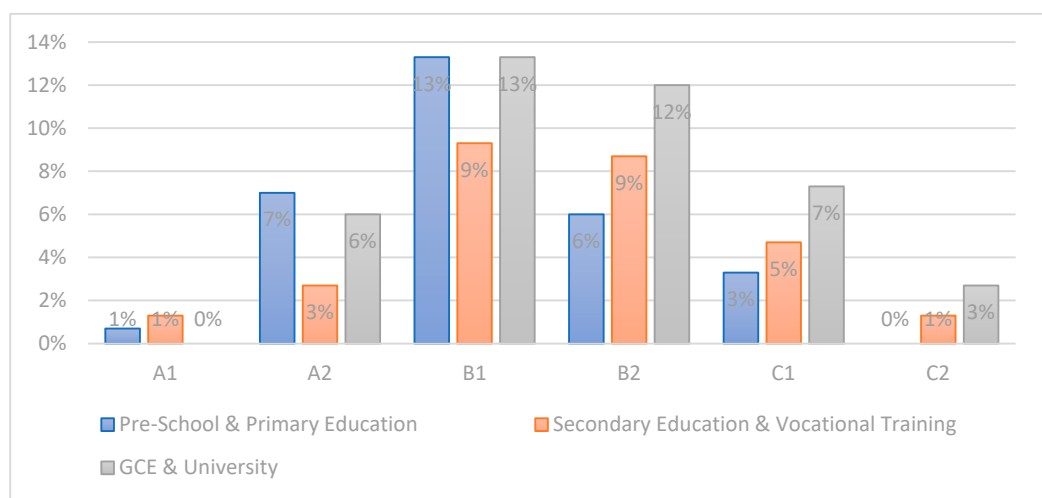

**Figure 1.** Distribution by levels of competence according to the educational stage.

As shown in Table 3, the means obtained in the different fields of competence show that those in which there is greater competence are those linked to the use of digital resources (3.32), student empowerment (3.24), professional commitment (3.19) and digital pedagogy (3.09); however, they show lower average values in the fields of assessment and feedback (2.99) and in the field related to facilitating students' digital competence (2.71), with the latter field of competence showing the lowest level of all those studied.

**Table 3.** Values obtained in relation to the fields of competence.

| Field of Competence | Mean | Standard Deviation |
|---|---|---|
| Professional commitment | 3.19 | 1.108 |
| Digital Resources | 3.32 | 1.195 |
| Digital Pedagogy | 3.09 | 1.341 |
| Assessment and Feedback | 2.99 | 1.366 |
| Empowering students | 3.24 | 1.487 |
| Facilitating Students' Digital Competence | 2.71 | 1.260 |

As seen in Table 4, there are differences between the different items that make up each field of competence. As for professional commitment, we find digital literacy is the item where the sample obtained the highest scores; however, within this section, a professional collaboration between teachers is the one that obtains lower average scores than the others. In terms of digital assets, the selection, creation, and modification of digital assets are the most highly rated items, while data management, sharing and protection show a lower mean in this area. As for digital pedagogy, acting as a guide is the section that obtains a higher score than the rest. In contrast, favouring self-directed learning is the least valued aspect within this competence area. In relation to assessment and feedback, all items show an even intermediate level, mainly highlighting feedback, participation, analysis and evidence of competence in assessment strategies. In terms of empowering students, encouraging active student participation is the element with the highest score, while establishing differentiation and personalisation of content for students is the lowest rated. Finally, in terms of facilitating students' digital competence, the average scores are slightly lower than in the other fields of competence, as it has two items with the lowest scores among all the elements that make up the different competence areas; digital

communication and collaboration (1.67) and responsible use and well-being (1.69); on the other hand, students' creation of digital content is the element with the highest mean within this area.

**Table 4.** Values obtained in relation to each item of the fields of competence.

| Field of Competence | Competence | Mean | Standard Deviation |
|---|---|---|---|
| Professional commitment | 1. Organisational communication | 2.39 | 0.834 |
| | 2. Professional collaboration | 1.99 | 1.013 |
| | 3. Reflective practice | 2.33 | 1.071 |
| | 4. Digital literacy | 2.87 | 1.162 |
| Digital resources | 1. Selection | 2.46 | 0.960 |
| | 2. Creation and modification | 2.55 | 0.952 |
| | 3. Management, sharing and protection | 2.14 | 1.170 |
| Digital pedagogy | 1. Teaching | 2.33 | 1.196 |
| | 2. Guidance | 2.38 | 1.309 |
| | 3. Collaborative learning | 2.43 | 1.114 |
| | 4. Self-directed learning | 1.93 | 1.191 |
| Assessment and feedback | 1. Assessment strategies | 2.05 | 0.979 |
| | 2. Analysis and evidence | 2.16 | 1.199 |
| | 3. Feedback and participation | 2.21 | 1.145 |
| Empowering students | 1. Accessibility and inclusion | 2.28 | 1.386 |
| | 2. Differentiation and personalisation | 2.11 | 1.396 |
| | 3. Active student participation | 2.55 | 0.994 |
| Facilitating students' digital competence | 1. Information and media literacy | 2.00 | 1.259 |
| | 2. Digital communication and collaboration | 1.67 | 1.157 |
| | 3. Creation of digital content | 2.28 | 1.205 |
| | 4. Responsible use and well-being | 1.69 | 1.118 |
| | 5. Digital problem solving | 1.91 | 1.161 |

### 3.1. Analysis of the Professional Commitment Sphere According to Educational Stage

With regard to the Professional Commitment sphere and its link to the educational stage in which the participating teachers work (Figure 2), as shown in Figure 2, there is a clear difference at the higher levels. In terms of the groups established for analysis, preschool and primary school teachers are reflected in the low (A1 and A2) or intermediate (B1 and B2) levels, with a higher presence in the A2 and B1 categories; however, they are not reflected in either of the two higher levels. On the other hand, secondary education and vocational training teachers are the only ones who score at level C2, indicating that they have a higher ability in this area, and they are also the most represented group in category A2; in terms of distribution, most of them fall into the intermediate levels (B1 and B2). Finally, the group of GCE and university teachers shows the highest percentage of people in the intermediate and high levels (B1, B2 and C1). However, they are not represented at level C2 and have participants at the lowest levels (A1 and A2).

### 3.2. Analysis of the Digital Resources Sphere According to the Educational Stage

If we look at the Digital Resources dimension (Figure 3), the scores are generally average, and the bulk of the sample accumulates around levels A2, B1 and B2. Looking in detail at the groups of teachers created, preschool and primary teachers are the most represented at the A1 level and are under-represented at the highest level, C2; most of these teachers are between A2 and B2 level, with a remarkable representation at the C1 level, second only to GCE and university teachers. The group of secondary education and vocational training teachers shows an equal representation at the highest proficiency levels (C1 and C2), with a large majority at the intermediate stages (B1 and B2) and a lower representation at A1 and A2. Finally, the group with the highest number of participants in

the higher stages (C1 and C2), generally framed in the intermediate and high levels, with very high values compared to the other participants in the B2 and C1 levels.

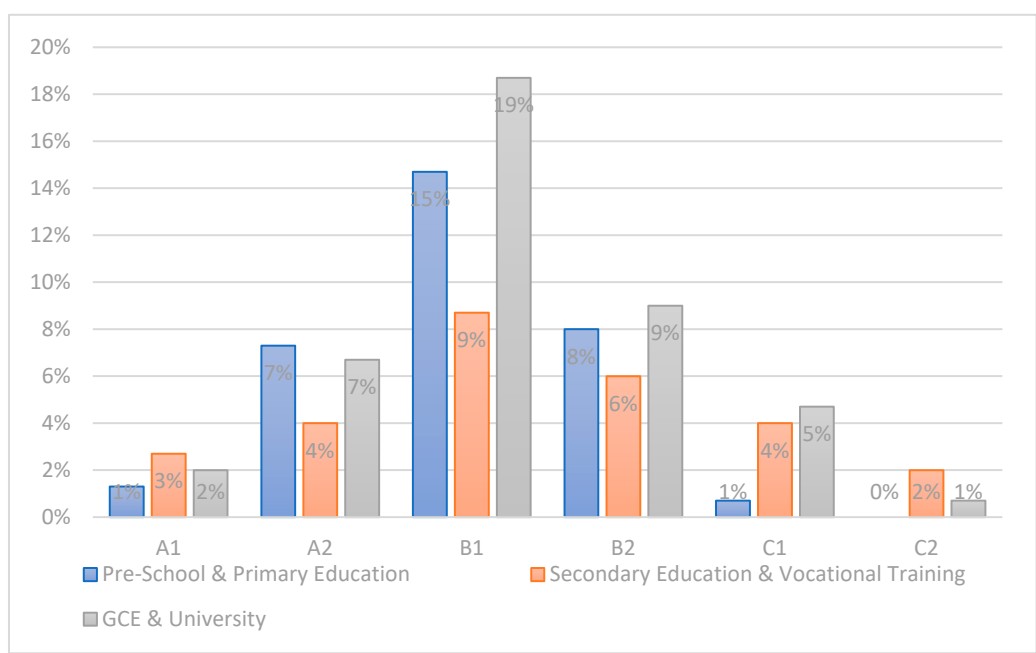

**Figure 2.** The competence level of Professional Commitment as a function of the educational stage.

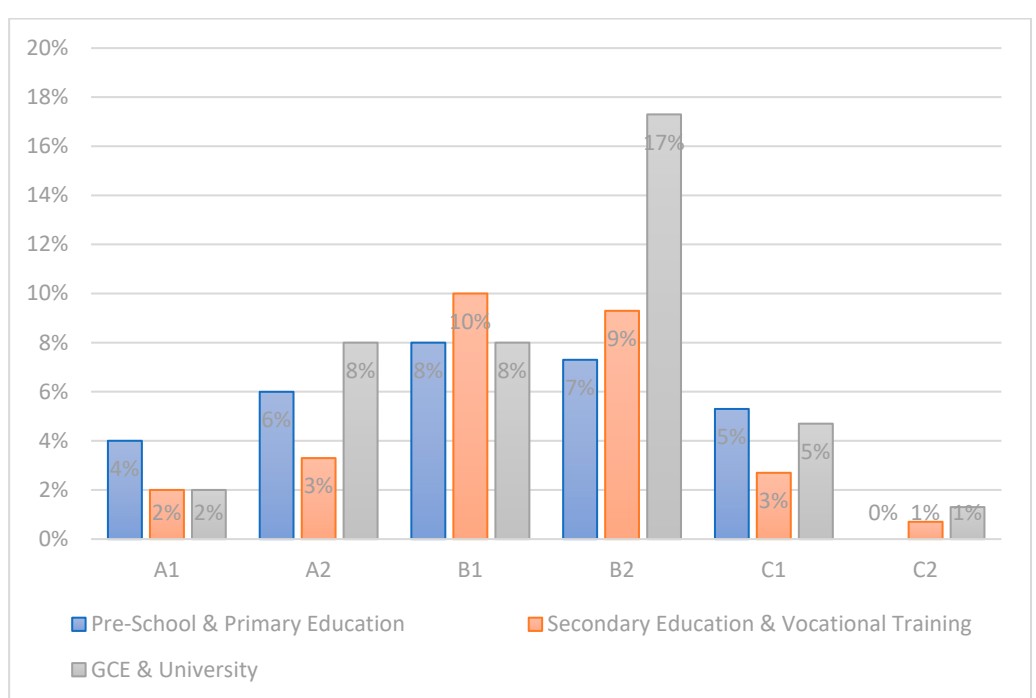

**Figure 3.** The competence level of Digital Resources as a function of the educational stage.

### 3.3. Analysis of the Digital Pedagogy Sphere According to the Educational Stage

As we can see in Figure 4, in relation to the Digital Pedagogy sphere according to the stage at which the teachers in the sample work, we find that preschool and primary school teachers are mainly at the lowest stages (A1 and A2), and show a small representation at B1 and B2 levels, as well as at C1; however, they are not represented at the highest level, C2. As for secondary school and vocational training, teachers are represented at all levels, with a notably higher representation in the intermediate stages (B1 and B2). They

also show higher results than secondary school and university teachers at the lower levels and lower at the higher stages; they are represented at all possible levels. Finally, with regard to the group of GCE and university teachers, they are the most represented in the intermediate and high levels (B1–C2) and the group with the least representation in the A1 and A2 stages.

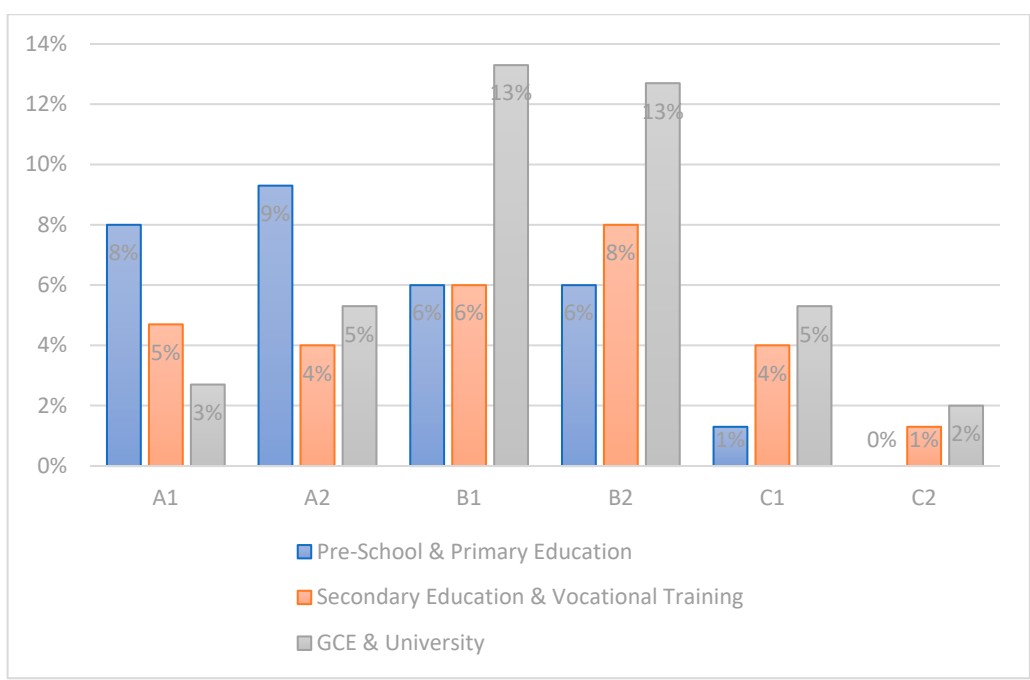

**Figure 4.** The competence level of Digital Pedagogy according to the educational stage.

*3.4. Analysis of the Assessment and Feedback Sphere According to Educational Stage*

Considering the dimension of Assessment and Feedback, in Figure 5, we find preschool and primary school teachers show very high values in the low and lower-intermediate stages, concentrating on these, A1, A2 and B1, most of the sample belonging to this group; however, they also provide representation at level B2 and C1, a level at which all groups are equally represented; however, they have no representatives at level C2. In general, secondary education and vocational training teachers have low values at levels A1 and B2, while the sample is concentrated at stages A2 and B1; they are also represented at higher levels, although to a lesser extent at the highest competence level, C2. With regard to GCE and university teachers, they are represented at all stages, with a more remarkable presence at levels A2, B1 and B2; in this group, it is worth noting that they are the most represented at level C2 and shows a greater presence than the rest of the groups in the intermediate stages (B1 and B2).

*3.5. Analysis of the Empowering Students Sphere According to the Educational Stage*

In relation to the Empowering Students sphere (Figure 6), preschool and primary teachers are mainly clustered around the intermediate and low levels. However, they are also represented at level C1, although no teachers in this group are linked to the highest competence level, C2. As for secondary education and vocational education, teachers are similarly distributed in the lower and intermediate levels, although their representation drops slightly in the higher stages, especially at the C2 level. Finally, GCE and university teachers show a higher level of competence, being the most represented in the intermediate and high levels; it is particularly striking that they are the group with the highest level of competence in this dimension and present the highest values in levels B1 and B2.

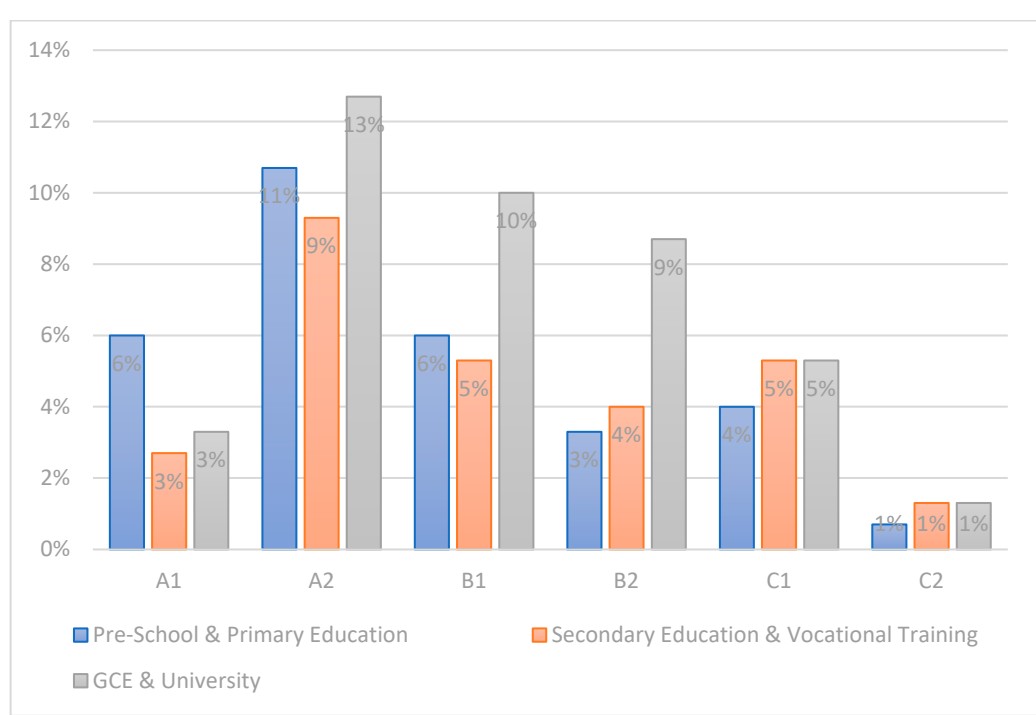

**Figure 5.** The competence level of Assessment and Feedback according to the educational stage.

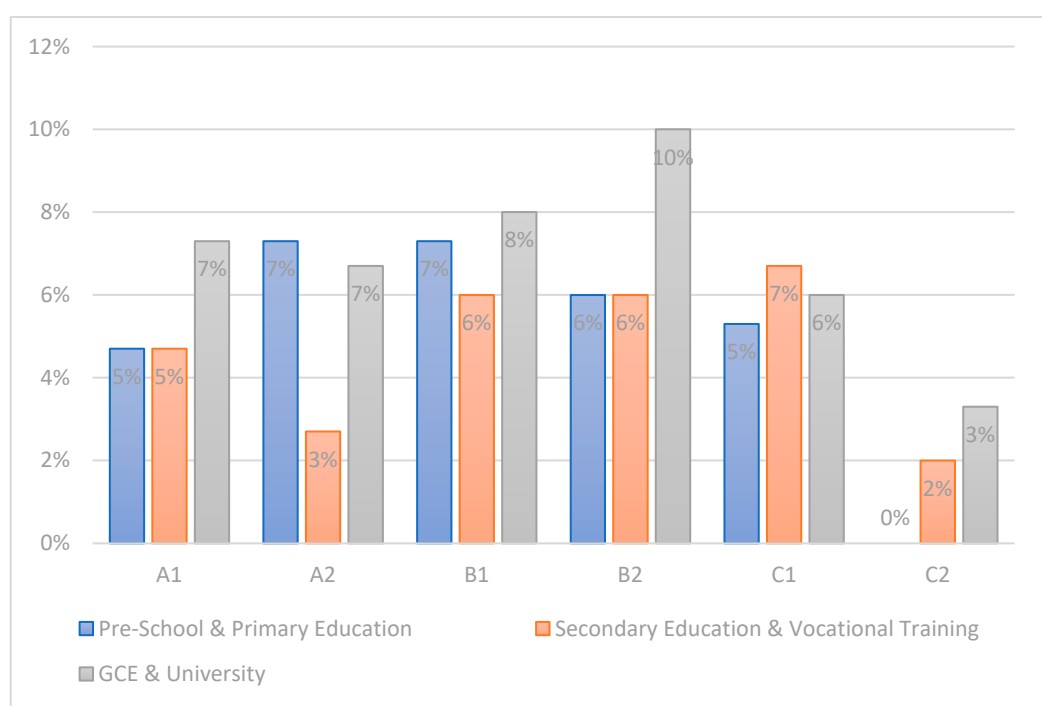

**Figure 6.** The competence level of Empowering Students according to the educational stage.

### 3.6. Analysis of Promoting the Digital Competence of Students Sphere According to Educational Stage

In terms of the proposed dimension, Figure 7 shows that preschool and primary school teachers have the lowest level of competence in terms of fostering students' digital competence, being the group, most represented at level A1; we also observe that they are not represented at stages C1 and C2, the highest level of competence. As for secondary education and vocational training teachers, they are not represented at level C1, although they are represented at level C2; on the other hand, most of this population is clustered in

the intermediate levels (B1 and B2), and they are under-represented at the stages associated with the lowest competence level. Finally, GCE and university teachers are more highly represented than the group of secondary and vocational teachers at levels A1 and A2, as well as at intermediate levels (B1 and B2); they are also the most represented group at the stages linked to the highest competence levels (C1 and C2).

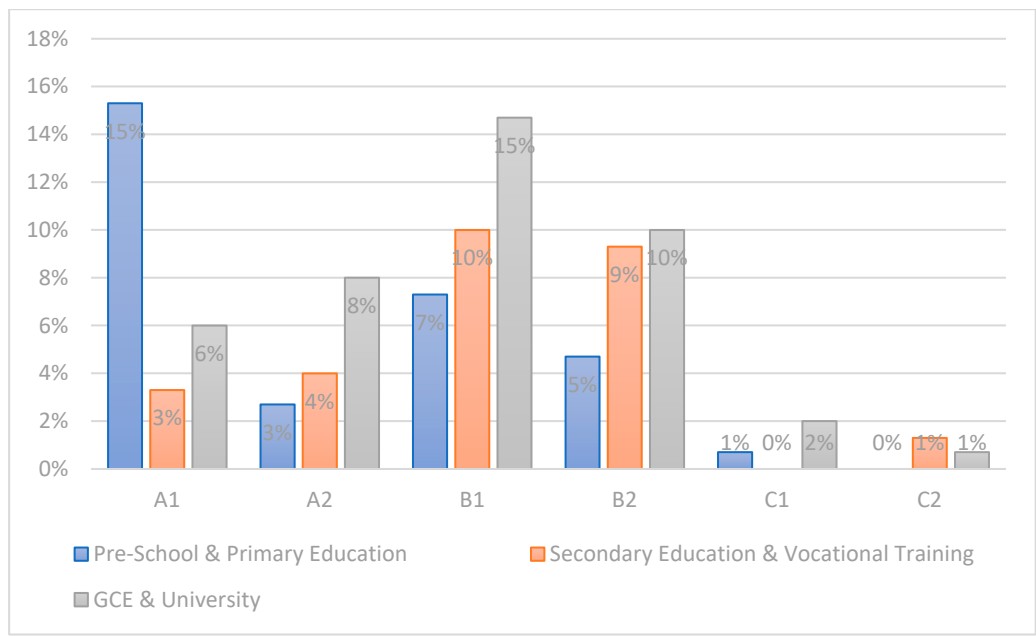

**Figure 7.** The competence level of Promoting the Digital Competence of Students sphere according to the educational stage.

## 4. Discussion and Conclusions

This study aims to describe the level of competence of teachers in Spain belonging to different educational stages, trying to identify the existing differences between them to obtain a globalised vision of the reality of those who make up the teaching staff at the different educational levels. It is also aimed at providing a broader picture of the differences between teachers at different stages of education. The importance of this subject lies in today's socio-cultural and educational reality, in which technologies are one of the main tools not only in the educational and work environment but have taken over all aspects of people's daily lives, substantially modifying how we communicate and interact; all these reasons being the main drivers of the change in teacher training needs.

Based on the pre-established levels in the questionnaire taken as a reference [5], it is clear that the teachers in the sample show an intermediate level, mainly collected between the levels (B1 and B2), with a small part of the sample in the pioneer and leader levels; there is also a small part located in the lower competence stages (A1 and A2), all of which is in line with previous studies such as those of Casal-Otero et al. [11], Torres-Barzabal et al. [6] and Hurtado-Mazeyra et al. [16] among others, where the participating teachers also have intermediate levels of competence and the extremes of competence are polarised by minorities, those who lack skills and abilities for the technologies, and those who have an excellent level of mastery. Likewise, and in line with the study carried out by Rodríguez-Hoyos et al. [17], it can be assumed that teachers, in relation to the teaching-learning processes, try to involve technologies to escape from traditional models and offer students a guiding thread that allows them to be participants in their learning. Let's look at studies carried out in other countries, such as that by Días-Trindade et al. [18], which assesses the competence level of primary and secondary school teachers in Portugal. It indicates that they have an average B1 level, as does the sample studied in the research presented here.

Considering the preschool and primary school teachers participating in the study, they have similar means to teachers in other studies at the same level, such as the one conducted by Hurtado-Mazeyra et al. [16], which shows that the great majority of them are framed within the integrating level (B1);. However, in this case, the results differ slightly since the next largest group is framed within level A2, followed by B2, in contrast to the reference study in which level B2 is the second largest. Likewise, this study and the reference study show the scarcity or absence of teachers at the most advanced level of competence (C1 and C2).

As for secondary education and vocational training teachers, taking as reference studies such as Casal-Otero et al. [11], the results coincide with the stage we refer to, although the sample studied shows slightly higher values than the reference sample. However, this may be due to the size of the population under study.

Finally, with regard to GCE and university teachers, the results coincide with the reference studies, mainly those conducted by Torres-Barzabal et al. [6], which show that university teachers have an intermediate level of competence. In that case, the sample indicates slightly higher scores in some of the sections analysed.

In conclusion, and considering the different phenomena that have taken place, digital competence is emerging as an imperative need in both in-service teacher training and in the training of future teachers since it is in their hands that media literacy and the training of children and adolescents in the future will be left so that they can become digitally competent. Therefore, we can affirm with certain categoricalness the need to emphasise this training, given that, based on what was established in 2022 in the Agreement of the Sectoral Conference on Education [19], which sets out the need to certify, accredit and recognise the digital competence of teachers, the need for teachers to possess the necessary skills and tools that enable them to be competent in digital matters and, at the same time, to be able to transmit this knowledge to train the students they teach is already a reality.

In terms of proposals for improvement and future avenues for research, it could be interesting to analyse the target population using other questionnaires or to try to carry out various tests that objectively measure teachers' real competence and not only their self-perceived competence, which would make it possible to compare the results and establish a relationship between obtained and perceived competence; for example by comparing the level assigned by the competent educational bodies for in-service teachers with the result obtained in the questionnaire on the self-perceived level of digital competence through the DigCompEdu Check-In questionnaire. These proposals would allow us to delve deeper into the educational reality of teachers in terms of their digital competence.

In general terms, we have found the imperative need for training in digital competencies. Although however, according to Jiménez-Hernández et al. [20], there is still a long way to go. The progress made has been remarkable, with teachers and learners becoming increasingly digitally literate, which bodes well for the future. To a large extent, this can be achieved by continuing to promote research and studies and adapting models of analysis of Teachers' Digital Competence to adapt them to the new realities arising from the scientific, technical and social developments of the future. Likewise, it is vitally important to consider the so-called soft skills, as Antón-Sancho et al. [21] explain in their study, given that they are necessary for the development of any type of training in the virtual learning environment and also favour the acquisition and development of digital competence in teaching.

Similarly, as Hidson [22] states, the political leaders of the different bodies linked to education must involve teachers and the needs they present to make decisions on the professional competencies they should possess and the implementation of these competencies, as well as the support they may require for their correct implementation. Furthermore, we must consider the reality in which we operate, a reality that may be conditioned by situations arising from health emergencies, such as COVID-19, which have substantially modified the role played by information and communication technologies; in this sense, studies such as that of Antón-Sancho et al. [23], highlight the growing importance of technology in all educational environments. To sum up, what has been presented so far through

the studies by Hidson [22] and Antón-Sancho et al. [23] points to some of the main keys to the future of education and its necessary adaptation to the reality in which it is produced.

With regard to the training needs of teachers in terms of digital competence, and to endorse the above, it should be noted that, as Betancur-Chicué and García-Valcárcel [24] state, teacher training plans must analyse and promote collaborative and autonomous learning in relation to information and communication technologies on the part of future teachers, effective communication through technology and the use of didactic tools specific to each area of knowledge; finally they also detect the need to highlight the potential of technologies for assessment purposes and the evident lack of teachers' ability to optimise the data they offer. Therefore, assessment plans must be designed based on these tools to offer the best possible feedback to students.

**Author Contributions:** Conceptualisation: M.Á.G.-D., S.R.-C., V.D.-B. and C.D.G.-V.; Investigation: M.Á.G.-D., S.R.-C., V.D.-B. and C.D.G.-V.; Resources: M.Á.G.-D., S.R.-C., V.D.-B. and C.D.G.-V.; Writing—original draft preparation: M.Á.G.-D.; Writing—review and editing: M.Á.G.-D., S.R.-C., V.D.-B. and C.D.G.-V. All authors have read and agreed to the published version of the manuscript.

**Funding:** This research received no external funding. It is part of the Doctoral Thesis writing by D. Miguel Ángel García–Delgado and directed by Dra. Sonia Rodríguez-Cano.

**Institutional Review Board Statement:** The study was conducted according to the guidelines of the Declaration of Helsinki and approved by the Ethics Committee of University of Burgos (IR 3/2023; date of approval: 7 February 2023).

**Informed Consent Statement:** Informed consent was obtained from all subjects involved in the study.

**Data Availability Statement:** Due to privacy and confidentiality issues the data are not available.

**Conflicts of Interest:** The authors declare no conflict of interest.

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
