# Peer review of "Digital Teaching Competence among Teachers of Different Educational Stages in Spain"

_education, doi:10.3390/educsci13060581_

Round 1
Reviewer 1 Report
The article under review explores the competence level of teachers in integrating new technologies into the teaching-learning process in the context of the new educational reality. The authors conducted a descriptive study involving 150 teachers from various educational stages in Spain. The study utilized the DigCompEdu Check-in questionnaire to assess the competence level of the teachers.
In terms of content description and contextualization, the article provides a succinct overview of the research topic within the current theoretical background. However, to enhance the comprehensiveness of the analysis, it would be beneficial for the authors to incorporate more literature from outside the Spanish academic realm. This would provide a broader perspective on the use of technology in education and allow for a more comprehensive understanding of the research topic.
The article includes relevant references that contribute to the research. The research design, questions, hypotheses, and methods are clearly stated, facilitating the understanding of the study's objectives and approach. The authors effectively present their arguments and discuss the findings in a coherent and balanced manner.
Regarding the presentation of empirical results, the article effectively communicates the findings of the study. The results indicate an intermediate level of competence among teachers at all educational stages surveyed, with variations observed depending on the specific stage of education. However, to further strengthen the study, the authors could consider incorporating additional references from research conducted in other countries. This would enable a comparison of the competence levels of teachers across different educational systems, providing valuable insights into the performance of teachers in various contexts.
The article meets the requirements of being adequately referenced, providing the necessary citations to support the arguments and findings. However, as previously mentioned, incorporating more literature from outside the Spanish academic realm would further enhance the article's depth and breadth. Additionally, the authors could provide a broader discussion on the implications of their findings for higher education, as the current conclusions are relatively brief.
In summary, the article successfully addresses the competence level of teachers in integrating new technologies into the teaching-learning process. To improve the article, it is recommended to incorporate more literature from outside the Spanish academic realm, specifically focusing on research conducted in other countries to provide a comprehensive analysis. Additionally, expanding the conclusions to encompass a broader perspective on higher education would further enhance the article's overall quality.
The level of academic English in the article is at the necessary level, allowing for clear and effective communication of the research results. The authors effectively convey their findings and ensure that the reader can understand and interpret the results without difficulty.
Author Response
First of all, we would like to thank you for taking the time to review the article, and as you rightly say, the aim is to carry out a descriptive study of teachers at different educational stages in order to establish a basis for determining the competence level of the participating teachers through the DigCompEdu Check-In questionnaire.
Furthermore, in order to provide a more extensive overview of the research topic, new international references have been included, which you can see in lines 366 to 369.
Thank you for your positive assessment of the way in which the research has been carried out and for the coherent results that facilitate the overall understanding of the article.
In order to obtain an overview and to facilitate comparison with different educational systems, we have included studies that refer to the competence level of teachers in Portugal (lines 366 to 369). In future lines of research we will carry out this research with Portuguese teachers.
As stated in his review, the implications of the findings for higher education have been extended, providing data from previous studies to improve the quality and extent of the current ones (lines 428 to 437).
Finally, and as explained above, the proposed modifications have been made, including several references outside the Spanish academic sphere, as well as the improvement and expansion of the conclusions in order to improve the quality of the article.
With respect to the level of English, we thank you for your comments in this respect and hope that this will favour the understanding and interpretation of the research described.

Reviewer 2 Report
Title and abstract
The title effectively conveys the study's topic, while the abstract provides a brief overview of the study's objective, methodology, key findings, and implications. However, the abstract could benefit from providing more details about the methodology, elaborating on the proposed improvements, and offering specific recommendations for training teachers in digital competence. These additions would enhance the clarity and completeness of the abstract.
Introduction
the introduction provides a solid background on the importance of digital competence for teachers in the context of technological advancements in education. However, it could be strengthened by clarifying the research objective and providing more context on the frameworks and assessments of digital competence that will be used in the study.
Methodology
the research design could benefit from a more detailed explanation of the sampling method employed. The passage mentions a sample size of 150 teachers from different educational stages but does not provide information about the sampling strategy or how representative the sample is of the broader population of teachers in Spain. Additionally, the passage lacks information about the data analysis methods that will be used to analyze and interpret the collected data.
Results
The provided exposition of results presents an analysis of Teacher Digital Competence grouped according to educational stages. It evaluates the level of competence in different areas and compares the performance of teachers in each stage. However, there are some critical points to consider regarding the scientific and coherent presentation of the results:
Lack of Methodological Details: A scientific analysis should include a clear description of the methodology employed to ensure transparency and replicability.
Lack of Comparative Analysis: While the exposition highlights differences in competence levels among educational stages, it does not provide a comparative analysis with other relevant factors such as years of teaching experience, professional development opportunities, or technological infrastructure. Without considering these variables, it becomes challenging to attribute the observed differences solely to educational stage. A scientific analysis should account for potential confounding factors and provide a more nuanced interpretation.
Discussion
The provided discussion describes a study aimed at assessing the level of competence of teachers in Spain across different educational stages, with a focus on the use of technology. The discussion provides an overview of the findings and compares them to previous studies, highlighting similarities and differences. It also emphasizes the importance of digital competence in teacher training and the need to promote further research and adaptation of models.
Overall, the discussion offers a scientific and coherent analysis of the study. However, there are a few points to consider in terms of a critical evaluation:
- Lack of Methodological Details: The discussion lacks specific details about the methodology employed in the study. It would be beneficial to know how the competence levels were measured, the sample size, and the criteria used to categorize the teachers into different stages. Without this information, it is difficult to fully assess the reliability and validity of the findings.
- Limited Scope: The discussion mainly focuses on the level of competence of teachers and the differences observed between educational stages. While this is a valuable aspect to explore, it would be beneficial to also consider other factors that may influence teachers' competence, such as training programs, teaching methods, and classroom practices. A more comprehensive analysis would provide a broader understanding of the factors affecting digital competence in teachers.
- Lack of Discussion on Implications: Although the discussion highlights the importance of digital competence in teacher training, it does not delve into the implications of the findings. How can the results be used to improve teacher training programs or inform policy decisions? Including a discussion on the practical implications and recommendations for improving digital competence among teachers would enhance the scientific value of the study.
- Self-Perceived vs Objective Competence: The discussion mentions the need to measure teachers' real competence rather than solely relying on self-perceived competence. This raises the question of whether the findings accurately reflect the teachers' actual digital competence or if they are influenced by their perception of their own abilities. Future studies could consider incorporating objective measures of competence to provide a more comprehensive assessment.
In conclusion, the discussion provides valuable insights into the level of digital competence among teachers in Spain. However, it would benefit from additional methodological details, a broader analysis of influencing factors, a discussion of implications, and a consideration of objective measures of competence. Addressing these points would strengthen the scientific rigor and coherency of the discussion.
Author Response
First of all, we would like to thank you for taking the time to review our work. As requested, the summary has been expanded with more methodological details (line 8), the proposed improvements have been included and some specific recommendations for teacher training in digital competence have been provided (lines 15 to 17).
With regard to the recommendations set out in the introduction, the aim of the research has been made explicit (line 82).
With regard to methodology, the type of sampling used (lines 157 to 161) and the type of analysis used to carry out the study (lines 157 to 161) have been included.
With regard to the results, the methodology used to analyse the data has been included (lines 168 to 171), and the factor of teaching experience has been added as a further characteristic to be taken into account (lines 171 to 182).
Based on the proposed recommendations, methodological details have been added regarding data collection and the criteria used to classify teachers according to their educational stage, an aspect which is included in the questionnaire and explained in line (170 , 171), based on the organisation of the Spanish educational system.
Likewise, the analyses and comparisons with previous studies have been extended (lines 400 to 403).
As for the lack of debate on the implications of the results for teacher training, some further nuances have been included (lines 428 to 437) as well as a debate on the practical implications and possible recommendations for improving the digital competence of teachers.
Finally, and as explained in lines 400 to 403 the need has been established to carry out a future study in which the similarity and differences between self-perceived and objective competence can be compared using different tools, in order to establish the suitability of the tool used and to find out whether the results are similar to the skills that teachers present in relation to this competence.
